# Changes in the Neurochemical Coding of the Anterior Pelvic Ganglion Neurons Supplying the Male Pig Urinary Bladder Trigone after One-Sided Axotomy of Their Nerve Fibers

**DOI:** 10.3390/ijms22052231

**Published:** 2021-02-24

**Authors:** Żaneta Listowska, Zenon Pidsudko

**Affiliations:** Department of Animal Anatomy, Faculty of Veterinary Medicine, University of Warmia and Mazury in Olsztyn, Oczapowskiego Street 13, 10-719 Olsztyn, Poland; zaneta.listowska@uwm.edu.pl

**Keywords:** plasticity of pelvic neurons, unilateral axotomy, anterior pelvic ganglion, immunohistochemistry, quantitative real-time PCR, neurotransmitters, male pig

## Abstract

The present study investigated the effect of unilateral axotomy of urinary bladder trigone (UBT)-projecting nerve fibers from the right anterior pelvic ganglion (APG) on changes in the chemical coding of their neuronal bodies. The study was performed using male pigs with immunohistochemistry and quantitative real-time PCR (qPCR). The animals were divided into a control (C), a morphological (MG) or a molecular biology group (MBG). APG neurons supplying UBT were revealed using the retrograde tracing technique with Fast Blue (FB). Unilateral axotomy resulted in an over 50% decrease in the number of FB+ neurons in both APG ganglia. Immunohistochemistry revealed significant changes in the chemical coding of FB+ cells only in the right ganglion: decreased expression of dopamine-B-hydroxylase (DBH)/tyrosine hydroxylase (TH) and up-regulation of the vesicular acetylcholine transporter (VAChT)/choline acetyltransferase (ChAT), galanin (GAL), vasoactive intestinal polypeptide (VIP) and brain nitric oxide synthase (bNOS). The qPCR results partly corresponded with immunofluorescence findings. In the APGs, genes for VAChT and ChAT, TH and DBH, VIP, and NOS were distinctly down-regulated, while the expression of GAL was up-regulated. Such data may be the basis for further studies concerning the plasticity of these ganglia under experimental or pathological conditions.

## 1. Introduction

The main source of autonomic innervation of the lower urinary tract of mammals is the pelvic plexus [1,2,3,4]. Its morphology is characterized by high interspecies diversity [2,3] ranging from a network containing numerous small ganglia in humans, pigs, rabbits, dogs and cats, to two clearly distinguished parts in the guinea pig: the cranial nerve network, with a large, paired ganglion called the anterior major pelvic ganglion (AMPG), and the caudal nerve network with numerous diffuse nerve ganglia [2]. In the male rat, most neurons form one large ganglion, called the major pelvic ganglion (MPG) and several additional (accessory) ganglia [2,3].

In the male domestic pig, very numerous pelvic neurons are organized in an orderly manner, and the largest and most cranial group of nerve cells are the anterior pelvic ganglia (APG) located on both sides of the bladder neck [1]. Pelvic ganglia are distinguished from other ganglia of the autonomic nervous system because they have a “mixed”, sympathetic–parasympathetic character [1,2,3,5]. Adrenergic neurons dominate in the cranial ganglia, while in the caudal ganglions they occur scarcely.

Pelvic neurons, in addition to classical neurotransmitters, i.e., noradrenaline (NA) and acetylcholine (ACh), express various neuropeptides, such as: vasoactive intestinal peptide (VIP), neuropeptide Y (NPY), galanin (GAL), somatostatin (SOM), substance P (SP), or calcitonin gene-related peptide (CGRP). They also show immunoreactivity to nitric oxide synthase (NOS) [1,2,4]. These biologically active substances may occur in cholinergic, adrenergic and non-adrenergic and non-cholinergic (NANC) neurons [6]. Key patterns of the chemical coding of pelvic neurons may differ significantly between species and sexes and, to a lesser extent, between races [2,4]. Pelvic ganglia show sexual dimorphism [3,5,6]. Male ganglia, compared to females, contain a much higher number of nerve cells and a greater percentage of them are adrenergic neurons [2,3,4].

Previous studies have shown the involvement of APG neurons in the autonomic innervation of boar urogenital organs. Fibers originating from the right and left APG supply the trigone and the neck of the bladder and are responsible for the innervation of the vas deferens, seminal glands and the prostate body [4]. 

Axotomy is a factor that leads to the degeneration and even death of nerve cells [7]. The disruption of the neurite brings to stop the retrograde transport of neurotrophic factors from peripheral tissues to the perikaryon, which is responsible for maintaining the basic functions of the cell body [7]. Nerve cells demonstrate numerous adaptation processes, enabling them to adapt to new conditions and to defend themselves against the effects of axotomy. The available data on the influence of axotomy on neurons of the autonomic system refer to cells located in the paravertebral ganglia (sympathetic trunk ganglia, including SChG, superior cervical ganglion-SCG, stellate ganglion) and prevertebral ganglia (MPG, CaMG, paracervical ganglion-PCG) [8,9,10,11,12,13,14,15,16,17,18,19]. Knowledge how pelvic neurons react to damage to their axons is fragmentary. The only available data concern changes in the chemical coding of CaMG and APG neurons supplying the testes in the boar after unilateral and bilateral orchidectomy [20,21].

The goal of this study was to create an experimental model to study the influence of damage to the autonomic nerve fibers supplying the urogenital organs on the biology of their neurons using the domestic pig. The domestic pig is the animal species most closely related to humans in anatomical, histological and physiological aspects [4,22,23]. Therefore, the experiment aimed to determine the effect of the unilateral axotomy of the nerve fibers projecting from the right APG to UBT on the number and distribution of UBT-projecting nerve cells in the right and left APG, the expression of selected biologically active substances in the right and left APG neurons (adrenergic neuronal markers—DBH, TH), cholinergic cell markers (VAChT, ChAT) and selected neuropeptides (VIP, NPY, GAL and NOS) at the level of protein and mRNA.

## 2. Results

### 2.1. Distribution and Number of FB+ Neurons in the APG Ganglia Supplying the UBT

In the control group, FB+ neurons were located in bilateral ganglia, and their total number was 16,539 (3308 ± 786.3 SEM). There was a difference in their number between the right and left APG. The right APG was characterized by a higher number of retrogradely labeled nerve cells compared to the left APG, amounting to 11,372 (2274 ± 575 SEM) and 5167 (1033 ± 312.2 SEM), respectively (Figure 1). Although the distribution of FB + neurons was uniform, in the right ganglion they were concentrated mainly in the area of the exit of the nerve fibers to the seminal vesicle and the urinary bladder trigone.

In the experimental group, as a result of the right-sided axotomy, there was a significant change in the number of FB+ neurons in both ganglia (Figure 1). Both in the right and left APG there was a decrease in their number. The total number of FB+ cells was 7679 (1536 ± 766 SEM) including 5415 in the right ganglion (1083 ± 607.2 SEM) and 2246 in the left ganglion (452.8 ± 162.2 SEM). In the right APG, FB + neurons were present mainly at the exit of nerve fibers to the seminal vesicle and UBT, while in the left APG they were concentrated mainly on one side, within the area supplied by the hypogastric nerve (Figure 1 and Figure 2a,b).

### 2.2. Immunohistochemical Characteristic of FB+ Neurons Projecting to the UBT

#### 2.2.1. Control Group

Immunohistochemical studies showed that FB+ neurons of the anterior pelvic ganglia formed three major neuronal populations. The vast majority of such cells in the right and left APG (60.39 ± 1.09% and 70.92 ± 0.84%, respectively) belonged to the group of adrenergic neurons (TH/DBH-IR). The second-largest population was cells belonging to the group of cholinergic neurons (ChAT/VAChT-IR), which accounted for 39.14 ± 0.94% of FB+ neurons on the right and 30.46 ± 1.16% on the left. The smallest population was made up of cells belonging to the group of NANC neurons (non-adrenergic/non-cholinergic), which made 1.64 ± 0.37% in the right ganglion, and 1.62 ± 0.29% of the total FB+ neuronal population in the left ganglion (Figure 3). Immunohistochemical staining also revealed that the FB+ neurons of both APG ganglia were immunoreactive for VIP (Figure 4a,c), NPY and GAL (Figure 5a,c), but did not express bNOS (Figure 6a,c and Figure 7a,c). Moreover, it was observed that the percentage of FB+ neurons containing the studied neuropeptides did not differ significantly between the right and left ganglia. NPY-IR cells accounted for 21.17 ± 0.49% of right and 22.28 ± 1.08% of the left APG retrogradely labeled neurons, while VIP-IR perikaryons accounted for 4.40 ± 0.19% and 4.69 ± 0.22%, and GAL 0.92 ± 0.03% and 0.98 ± 0.02%, respectively (Figure 3).

#### 2.2.2. Experimental Group

In the experimental group, it was observed that the unilateral axotomy significantly changed the chemical coding of FB+ neurons in the right APG (Figure 8). Immunohistochemical studies showed that after cutting the nerve fibers there was a drastic decrease in the number of FB+/TH/DBH-IR cells (from 60.39 ± 1.09% to 34.40 ± 2.43%) and a slight increase in the number of FB+/VAChT/ChAT-IR neurons (from 39.14 ± 0.94% to 48.00 ± 2.81%) as well as of NANC neurons (from 1.64 ± 0.37% to 19.49 ± 1.72%) in the right ganglion. Additionally, there was a significant increase in the number of VIP-immunoreactive FB+ nerve cells (from 4.40 ± 0.19% to 37.39 ± 1.13%) (Figure 4e,g) and GAL+ neurons (from 0.92 ± 0.03% to 36.63 ± 1.21%) (Figure 5e,g) and the presence of FB+ neurons expressing bNOS (18.38 ± 1.28%) was found (Figure 6e,g and Figure 7e,g). The percentage of FB+/NPY-IR neurons was statistically similar in the control and experimental groups and amounted to 21.17 ± 0.49% and 19.36 ± 1.28%, respectively (Figure 8 and Figure 9).

On the basis of the performed immunohistochemical staining, it was also found that the right-sided axotomy did not cause significant changes in the chemical coding of FB+ neurons in the left APG (Figure 10). FB+/TH/DBH-IR cells accounted for 69.49 ± 1.14% of the total left APG retrogradely labeled neurons. FB+/VAChT/ChAT-IR perikaryons made up 31.54 ± 1.21% and NANC neurons made up 1.72 ± 0.29% of FB+ neuronal population. In total, 21.48 ± 0.97% of FB+ neurons were immunoreactive for NPY, 3.49 ± 0.09% for VIP and 1.42 ± 0.23% for GAL. As in the control group, no cells expressing bNOS were found (Figure 9).

### 2.3. Distribution and Immunohistochemical Characteristics of Intraganglionic Nerve Fibers

#### 2.3.1. Control Group

Immunohistochemical staining showed numerous (++++) VAChT-positive intraganglionic nerve fibers of the right and left APG (Figure 7b). They formed bundles which, penetrating deep into the ganglia, separated the clusters of adrenergic and cholinergic neurons. Smooth VAChT-IR fibers, characterized by a longitudinal course, were located outside the clusters of nerve cells, in the peripheral regions of APG. It was observed that only single VAChT-immunoreactive fibers also contained VIP and NPY immunoreactivity. ChAT-IR fibers appeared as single (+), and their distribution was uniform over the area of the ganglia. They were visible as bands of smooth fibers running between nerve cells, and only a few showed the presence of GAL. In the middle part of the ganglia, scarce (++) varicose VIP-IR fibers (Figure 4c and Figure 11c) were observed, which were located between the FB+/TH/DBH-IR and FB+/VAChT/ChAT-IR perikaryons. The examined ganglia also showed a small (++) number of evenly spaced varicose NPY-IR fibers (Figure 8b). Most of them surrounded both adrenergic and cholinergic cells, and individual fibers divided the ganglia into smaller regions. No bNOS- and TH-immunoreactive fibers were found in either the right or left APG, although single (+) and unevenly distributed varicose GAL-IR fibers (Figure 12c) and DBH-IR were observed.

#### 2.3.2. Experimental Group

After axotomy, it was found that there was a significant increase in the density of DBH-IR and TH-IR fibers in both the right and left APG, while the number of VAChT-IR fibers decreased dramatically. The DBH-IR fibers were found in large numbers (+++) (Figure 4f and Figure 6f) and were arranged evenly throughout the cross-sections of the examined ganglia. They mainly supplied DBH-positive neurons and were part of the bundles of fibers that divided the ganglia into smaller regions. The vast majority of DBH-IR fibers showed immunoreactivity to NPY, and only a small number of them were VIP-IR (Figure 4h). The intraganglionic TH-IR fibers were numerous (+++) and unevenly distributed throughout the right and left APG. Most often they were concentrated around TH-positive neurons, forming numerous “basket-like structures”. Only some of the fibers surrounding TH-IR cells showed immunoreactivity to GAL. It was also found that a small number of TH-IR fibers were clustered in the caudal pole of the ganglia and they were largely simultaneously GAL-positive. In the experimental group, it was observed that in the right and left APG there are evenly distributed single VAChT-IR fibers belonging to the varicose fiber population. Only a few of them showed immunoreactivity to VIP and NPY. After unilateral axotomy, a significant increase in the density of VIP-IR (Figure 4g and Figure 10g) and GAL-IR (Figure 5g and Figure 11g) fibers was noted in the right and left APG. VIP-IR and GAL-IR fibers were evenly distributed, however, in the right ganglia they were slightly more numerous (++++) than in the left ganglia (+++). The vast majority of VIP-IR fibers were varicose fibers that formed “basket-like structures” around VIP-containing neurons. The varicose GAL-IR fibers most frequently entwined the GAL-containing cells and formed bundles separating cholinergic cells from adrenergic cells. In the experimental group, an increase in NPY-IR fiber density was noted only in the right APG (+++). In the left ganglion, their number was not changed (++). Most often they were visible as varicose fibers, mostly surrounding DßH-IR cells and forming basket-like structures around NPY-IR neurons. Single, smooth NOS-IR fibers were observed only in the terminal part of the right APG, while they were absent in the left ganglion.

### 2.4. Expression of Genes Coding for the Synthesis of Selected Biologically Active Substances in the Right and Left APG

The qPCR study showed that, as a result of the right-sided axotomy, the expression of genes responsible for the synthesis of biologically active substances in both APG ganglia changed. Compared to the control group, gene expression for VAChT and ChAT decreased, respectively, 3.95-fold and 21.46-fold in the right ganglion, and 2.88-fold and 4.82-fold in the left ganglion (Figure 13a,b). It was also observed that the expression of genes encoding DBH and TH decreased and was, respectively, 10.25 times and 4.72 times lower in the right APG and 5.53 times and 5.90 times lower in the left APG than in the control group (Figure 13a,b).

Moreover, it was found that the expression of the NOS gene showed a 14.80-fold decrease in the right and a 7.57-fold decrease in the left APG compared to the control group (Figure 13a,b). It was also observed that, after axotomy, VIP gene expression was 12.06 times lower in the right APG and 3.54 times lower in the left APG than in the control (Figure 13a,b). Moreover, in the right APG there was a 34.18-fold increase in the expression of genes for GAL, while in the left ganglion no statistically significant changes were observed in relation to the control group (Figure 13a,b).

In the right APG, the expression of the gene encoding NPY was 1.61 times higher than in the control, but there were no statistically significant changes in the expression of this gene in the left ganglion (Figure 13a,b).

## 3. Discussion

It was observed that the neurons projecting to UBT were located in bilateral ganglia, and their number was significantly higher in the right APG than in the left APG and it amounted to approx. 68.76% and approx. 31.24% of all the retrogradely-labeled nerve cells, respectively. Based on immunohistochemical stainings, it was found that FB+ neurons of bilateral APG ganglia belonged to three neuronal populations: adrenergic, cholinergic and NANC. They also showed immunoreactivity to VIP, NPY and GAL, but not to bNOS. The above results are largely in line with the observations made previously in the same animal species by other researchers [4,24]. It is well known that autonomic neurons may respond to damage to their nerve projections through numerous morphological and functional changes [3]. The cutting of nerve fibers deprives the perikaryons of the neurotrophic factors responsible for maintaining their basic life functions, which results in the activation of processes that may lead to the degeneration and even apoptosis of the neuron [7,11,25,26]. The experiment described in this paper showed that the right-sided axotomy procedure changed the number of FB+ cells both in the ipsilateral and contralateral APG. It was found that more than 50% of the FB+ neurons in the left and right APG died as a result of unilateral axotomy. The obtained results indicate that the pathological factor (the cutting of the autonomic nerve fibers) can induce extensive death of nerve cells both in the directly denervated and adjacent neurons. Different results were obtained in sexually immature gilts subjected to bilateral axotomy of the posterior colic nerves (*nervi colici caudales*) projecting from CaMG to the colon. In this experiment, the number of FB+ neurons in both CaMG ganglia did not change [11]. On the other hand, in rats, it was shown that cutting the sublingual nerve (*nervus hypoglossus*) resulted in the death of approx. 30% of the motor neurons, while approx. 70% of them survive [2,27]. Therefore, it seems likely that the survival of nerve cells after damage to their axons depends on the intrinsic properties of perikaryons, as well as the properties of the surrounding cellular environment and the type of tissues they supply.

The authors’ immunohistochemical studies showed that the right-sided axotomy significantly changed the chemical coding of right APG FB+ neurons, but did not cause significant changes in the retrogradely-labeled cells of the left ganglion.

In the experimental group, as a result of a unilateral axotomy, it was observed that the number of FB+/TH/DBH-IR neurons in the right APG decreased drastically from approx. 60% to approx. 34%. Similar results were obtained in studies of unilateral and bilateral orchidectomy in sexually mature boars. It was shown that both in animals undergoing right-sided and bilateral castration, the number of TH-IR neurons located in APG and CaMG decreased from approx. 60% to approx. 25–30%, and DBH-IR neurons from approx. 60 to approx. 50% [20]. A strong reduction of TH-immunoreactive neurons was also noted in CaMG in gilts undergoing bilateral axotomy of posterior colonic nerves (from approx. 87% to 61%), as well as following the partial or complete removal of the uterus [11,19]. These observations correspond with the changes noted in rat SCG neurons after cutting their axons [18]. It is believed that most or all pelvic adrenergic neurons are under the influence of nerve growth factor (NGF) which is synthesized in the tissues of the urogenital system [2]. Moreover, it is a widely accepted theory that the reduction of TH synthesis in axotomy-affected neurons derives from their deprivation of the neurotrophic factors, particularly NGF [8,21,28,29,30,31].

It is interesting that after unilateral axotomy, a large number of TH-IR fibers (from-to +++) appeared and the number of DBH-IR fibers increased (from + to +++) in the right APG. This phenomenon is difficult to explain at the moment. It may be presumed that the appearance of intraganglionic adrenergic fibers is a consequence of compensatory mechanisms developed in response to the massive loss of FB+/TH/DBH-IR neurons.

In the course of this study, it was also observed that after unilateral axotomy there was a drastic decrease in the number of VAChT-IR fibers (from ++++ to +) and a significant decrease in the expression of the genes encoding VAChT and ChAT in the right APG, while the number of VAChT/ChAT-IR neurons increased slightly (by approx. 9%). In the available literature, there are only reports on a transient decrease in both the number of VAChT-IR of nerve endings and the level of ChAT mRNA as a result of unilateral sublingual nerve axotomy in the rat [32,33,34].

It is generally accepted that one of the most important and distinctive properties of axotomy-affected neurons is their ability to alter their chemical coding pattern [9,12,28,35]. The current study showed that the right APG from experimental animals showed a significant increase in the number of FB+/VIP-IR and FB+/GAL-IR cells, while the number of FB+/NPY-IR cells did not change significantly. Moreover, it was observed that approximately 18% of the retrogradely-labeled neurons started to show immunoreactivity to bNOS. An increase in VIP and GAL expression was also found in APG and CaMG neurons of boars after unilateral and bilateral orchidectomy [20]. In previous studies, it was also observed that axotomy causes a decrease in NPY expression, but does not affect the expression of VIP and NOS in CaMG neurons in gilts, while it induces the synthesis of these substances in rat SCG [11,12,13,17]. The above-cited data support the hypothesis that the changes in neuropeptide expression in axotomy-affected neurons appear to be organ- and species-specific [11].

Currently, it is believed that biologically active substances such as GAL, VIP and NOS are crucial for the survival of neurons and the regeneration process of nerve endings [36]. In this experiment, it was observed that the increase in the number of FB+ neurons immunoreactive to GAL (from approx. 1% to approx. 36%) was accompanied by a significant increase in the number of GAL-IR fibers (from + to ++++), which was also correlated with a significant (34-fold) increase in gene expression of this neuropeptide. It is believed that overexpression of galanin is most likely the result of the action of leukemia inhibitory factor (LIF), which is released by damaged neurolemmocytes [17,28,37]. Moreover, it is believed that the increase in VIP expression following the axotomy may also be the result of perikaryon exposure to LIF, as well as the effect of depriving them of the NGF supply [17]. In this study, it was found that in response to the unilateral axotomy there was a significant increase in the number of FB+/VIP-IR cells (from approx. 4 to approx. 37%) and VIP-IR fibers (from ++ to ++++) in the right APG, while the expression of the VIP gene was significantly decreased (12-fold). It should be noted that the increase in the number of nerve cells immunoreactive to a given neuropeptide does not always correlate with an increase in the expression of their genes. It can be presumed that the reason for the decrease in VIP gene expression was the termination of the increased protein synthesis, which, in turn, was manifested by a significant increase in the number of VIP-IR cells and fibers compared to the control group. To clarify this hypothesis it is necessary to do further studies determining the change in the expression of VIP at the gene level versus the number of neurons and VIP-IR fibers versus the time elapsed from the autonomic fiber axotomy.

It is noteworthy that after cutting the pelvic plexus fibers, the presence of FB+/bNOS-IR neurons was found in the right APG, which, in turn, was not present in the control group. The qPCR study showed that the expression of the bNOS gene was higher in the control group than in the experimental group. A similar observation was made in the case of the expression of the NPY gene. It was found that following the axotomy the number of FB+ neurons, showing NPY immunoreactivity did not change significantly, while the expression of the gene encoding this neuropeptide decreased significantly compared to the control group.

The axotomy-induced increase in the expression of GAL, VIP and bNOS in the right APG neurons implies that these substances may participate in the processes related to regeneration and increase the viability of pelvic neurons.

The current study showed that right-sided axotomy did not significantly change the chemical coding of FB+ perikaryons in the left APG, but induced changes in the level of expression of genes coding for the synthesis of most of the studied substances. It also affected the number and neurochemical coding of intraganglionic nerve fibers which were comparable to the changes noted in the right ganglion.

## 4. Materials and Methods

### 4.1. Experimental Animals

The study was conducted on 20 sexually immature boars of the Large White Polish breed (8 weeks old and weighing approx. 10–15 kg) from a commercial pig fattening farm. The pigs were kept in groups (*n* = 5) in cages located in ventilated rooms with appropriate temperature and humidity, under natural daylight. The animals were isolated from noise and had free access to water and feed. All boars were subjected to experimental treatments approved by the Local Ethical Committee (license no. 36/2016) affiliated with the National Ethics Commission for Animal Experimentation, Polish Ministry of Science and Higher Education. The animals were divided into two experimental groups, each consisting of ten animals: the MG-group for morphological research and the MBG-group for molecular biology research. Each of these groups was divided into two subgroups: a control group (MG-C, MBG-C) and an experimental group (MG-E, MBG-E) consisting of five pigs.

### 4.2. Surgical Procedures and Injection of a Neural Retrograde Tracer

The animals fasted for 18 h before the planned surgery. All surgical procedures were performed under general anesthesia according to the following scheme: first, the boars were premedicated with atropine (Polfa, Poland; 0.04 mg/kg BW, SC), and azaperone (Stresnil, Janssen, Belgium; 2.5 mg/kg BW, IM). After approx. 15 min an intramuscular injection of ketamine was performed (Ketamine 10%, Biowet, Poland; 10 mg/kg BW) for the induction of general anesthesia and propofol then was administered after 20 min into the ear marginal vein (Scanofol, Scanvet, Poland; 4 mg/kg BW). During the operation, heart rate and respiration were constantly monitored.

### 4.3. Animals Intended for Morphological Research

All animals from the morphological group (MG, *n* = 10) underwent midline laparotomy. The abdominal incision was made in the white line from the umbilicus to the pubic symphysis and the urinary bladder was exposed. Subsequently, a total of 16 µL of 5% Fast Blue neuronal tracer (FB; Dr. Illing, Gross-Umstadt, Germany) suspension was injected under the serosa of the bladder trigone using a Hamilton syringe fitted with a 26-G needle. Eight injections of 2 µL of FB suspension were performed within the right and left side of the UBT, keeping a similar distance between the injections. To avoid leakage of tracer solution from the injection injury, the needle was left at the injection site for several seconds. The wall of the injected organ was then rinsed with physiological saline and gently wiped with gauze. This procedure aimed to avoid contamination of the adjacent tissue structures with the injected dye, which could adversely affect the reliability of the obtained results (other neurons that do not supply UBT could be labeled). After three weeks, all animals in the morphological group (MG, *n* = 10) were re-operated and midline laparotomy according to the procedure described above was performed. In the boars from the experimental subgroup (MG-E, *n* = 5), the aim of the reoperation was the right axotomy of nerve fibers from the APG ganglion to the urinary bladder trigone. This procedure was performed in the same way for each animal. Nerves from the right anterior pelvic ganglion were cut halfway along their route towards the UBT. In animals from the control subgroup (MG-C, *n* = 5), only manual manipulation was performed in the right part of the pelvic plexus, without cutting the nerve fibers. This consisted of removing it from the peritoneal cavity and exposing it for a few seconds.

### 4.4. Animals Intended for Molecular Research

Boars from the molecular biology group (MBG, *n* = 10) were used for studies aimed at determining changes in the expression of genes coding selected biologically active substances using the RT-qPCR technique (quantitative PCR reaction preceded by reverse transcription). After induction of general anesthesia, the animals underwent midline laparotomy (as previously described) but, unlike the morphological group animals, they were not injected with the Fast Blue neural tracer. In pigs from the experimental subgroup (MBG-E, *n* = 5), the right axotomy of nerve fibers projecting from the right APG ganglion to UBT was performed (Figure 3). In contrast, in the boars from the control subgroup (MBG-C, *n* = 5) only manipulations of the right pelvic plexus were performed without cutting the nerves (as described in the MG-C subgroup).

### 4.5. Collection, Fixation and Preparation of Tissues for Analysis

One week after the surgery, all boars from both experimental groups (MG, *n* = 10 and MBG, *n* = 10) were euthanized. They were anesthetized in accordance with the previously described general anesthesia, and the overdose of propofol led to breathing arrest. When the respiratory action ceased, in the boars from the morphological group (MG, *n* = 10) the thoracic cavity was opened by cutting the sternum in the midline. A metal cannula was then inserted into the left ventricle of the heart through which the fixative fluid (4% paraformaldehyde solution in 0.1 M phosphate buffer pH 7.4) was infused. At the same time, the auricle of the right atrium was cut off to allow the outflow of blood from the blood vessels.

Abdominal cavities were opened immediately after transcardial perfusion was completed and the right and left APGs were collected. The obtained tissues were additionally fixed by immersing them in a 4% paraformaldehyde solution for about 20 min and then washed several times in a phosphate buffer (pH 7.4). Subsequently, the collected material was placed in an 18% buffered sucrose solution (pH 7.4) with the addition of sodium azide and stored under refrigerated conditions (temperature 4 °C) until the ganglia dropped to the bottom of the container.

After the tissues were impregnated with sucrose, the collected ganglia were frozen and sliced with a cryostat (Leica CM1860) into 16 µm-thick sections. The sections were applied to pre-labelled chromalum-coated glass slides. They were then were dried for about 20 min at room temperature, placed in sealed boxes and stored at −20 °C. All the sections containing FB-labelled nerve cells were processed for double-labelling immunofluorescence with antibodies listed in Table 1 and the immunolabelling techniques were applied as described previously [38].

Standard controls, i.e., preabsorption for the neuropeptide antisera (20 μg of appropriate antigen per 1 mL of the corresponding antibody at working dilution; all antigens purchased from Peninsula, Sigma or Dianova) and the omission and replacement of all primary antisera by non-immune sera were applied to test antibody and method specificity.

The immunostained sections were studied and photographed with a Zeiss Axiophot fluorescence microscope equipped with epi-illumination and an appropriate filter set for FITC, Alexa Fluor 488, 555 i 568 and FB, and with a confocal microscope (Zeiss LSM 710). The relationships between immunohistochemical staining and FB distribution were examined directly by interchanging filters. The sections originated from different representative regions of the ganglion (from one of three different ganglion levels—upper, middle, and lower one-third). To determine the percentage of particular neuronal populations, at least 300 FB+ (FB-labelled) neuronal profiles were investigated for each combination of antisera. All FB+ cells found in particular sections were counted. To avoid double-counting the same neurons, the neuronal cells were counted in every fourth section. The number of immunolabelled profiles was calculated as a percentage of neurons immunoreactive to a particular antigen related to all FB+ perikarya counted. Finally, data were pooled from all animals in particular groups, expressed as means ± SEM and analyzed with GraphPad Prism 8 software with a paired Student’s *t*-test. The differences for which the level of significance was lower than 0.05 (*p* < 0.05) were considered statistically significant.

### 4.6. Molecular Research

After respiratory and cardiac arrest, the boars from the molecular group (MBG, *n* = 10) had their abdominal cavities opened and the right and left APGs were removed. The collected tissues were immediately placed in sterile tubes filled with RNAlater^®^ protection buffer (Qiagen, Düsseldorf, Germany) and stored at −20 °C. Total RNA was extracted using Total RNA Mini isolation kit (AA Biotechnology, Gdynia, Poland) and the cDNA samples were synthesized from respective high-quality matrix samples using Maxima First Strand cDNA Synthesis Kit for RT-qPCR (Thermo Scientific, Waltham, MA, USA). Quantitative real-time PCR was performed according to the method described by Kasica-Jarosz et al. (2018) [39] using SYBR Green (SYBR Select Master Mix, Applied Biosystems, Foster City, CA, USA) on 7500 Fast Real-Time PCR instrument (Applied Biosystems, Foster City, CA, USA). Oligonucleotide primers were designed using PrimerBLAST tool to detect gene expression of chosen markers [VACHT (slc18a1), ChAT (chat), DBH (dbh), TH (th), bNOS (N-nos), GAL (gal), VIP (vip), NPY (npy) and GAPDH (gapdh))]. Initial validation of reference genes revealed that for the purpose of the study, GAPDH showed the most efficient and equal expression among the samples. The values of the expression of the studied genes were calculated in each group as normalized to GAPDH expression. Each sample was analyzed in triplicate. The details are listed in Table 2.

The obtained results were analyzed with GraphPad Prism 8.0 (GraphPad Software Inc., San Diego, CA, USA) using the Student’s *t*-test for normally distributed data or using the Mann–Whitney test for data not meeting the assumptions of the normal distribution. The level of significance equal to 0.05 was adopted as statistically significant (*p* = 0.05).

## 5. Conclusions

This section is not mandatory but can be added to the manuscript if the discussion is unusually long or complex.

## Figures and Tables

**Figure 1 ijms-22-02231-f001:**
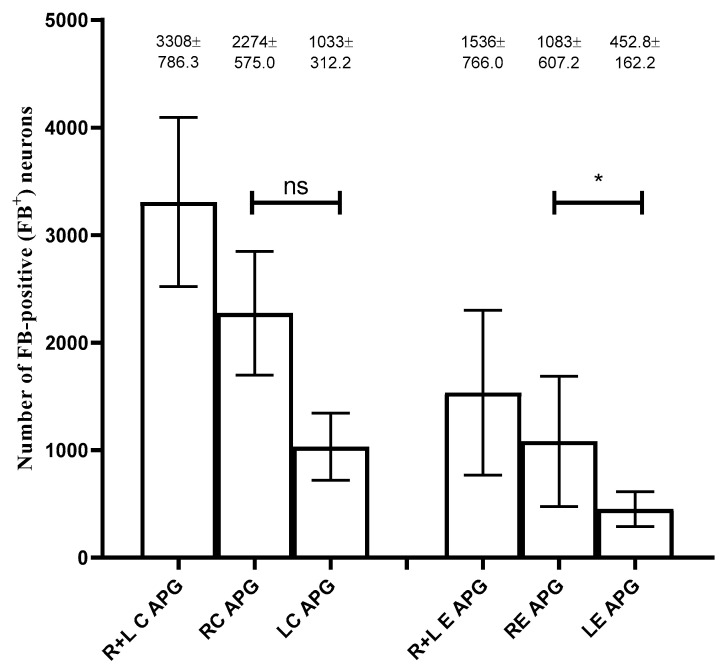
Total number of Fast Blue-positive (FB+) neurons (±SEM) in the anterior pelvic ganglia- anterior pelvic ganglia (APG) before-control (C) and after-experimental (E) unilateral axotomy. Bar diagram showing the distribution of the urinary bladder-projecting neurons in the left (L), C and E and right (R) C and E APG. Numerical data are given above the bars; “*”,-differences significant at *p* ≤ 0.05.

**Figure 2 ijms-22-02231-f002:**
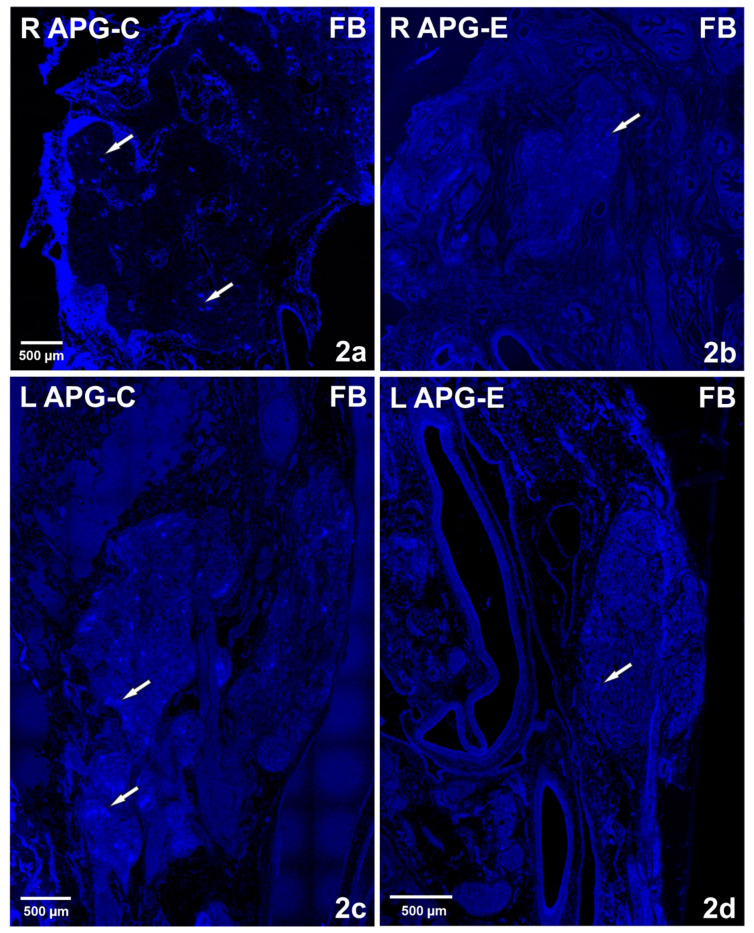
(**a**,**b**) Confocal laser scanning microscope images showing the distribution of FB+ neurons (in the right (R) APG from control (C) (**a**) and experimental (E) (**b**) animals. Significant reduction in the number of FB cells after unilateral axotomy surgery (**b**). Arrows indicate FB+ neurons. Scale bar = 500 µm. (**c**,**d**) Confocal laser scanning microscope images showing the distribution of FB + neurons in the left (L) APG from control (**c**) and experimental (**d**) animals. Marked loss of FB+ cells after right axotomy (**d**). Arrows indicate FB + neurons. Scale bar = 500 µm.

**Figure 3 ijms-22-02231-f003:**
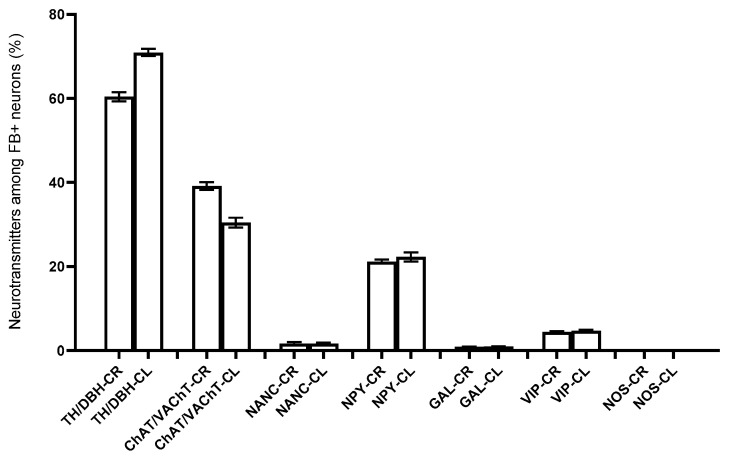
Percentage of FB+ neurons containing particular substances and neuropeptides located in the right (R) and left (L) APG of the control animals (C); There were no non-statistically (ns) significant differences (*p* > 0.05).

**Figure 4 ijms-22-02231-f004:**
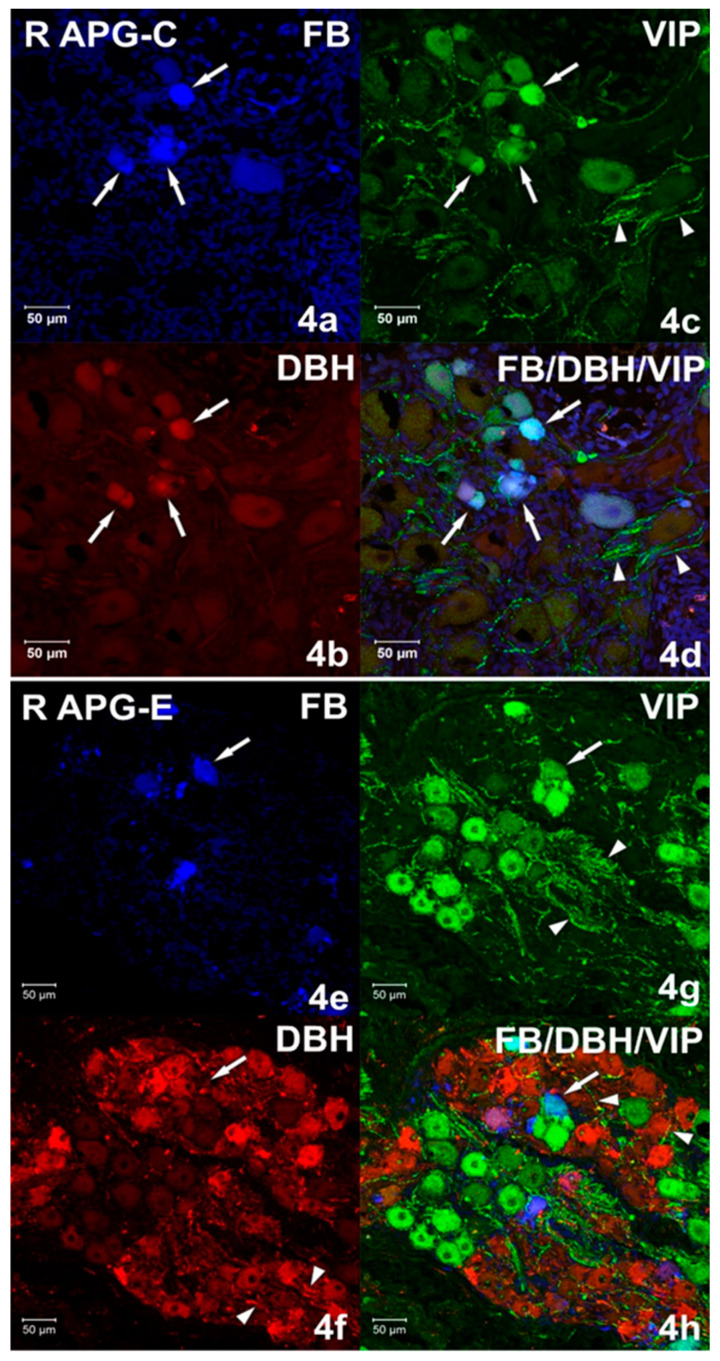
(**a**–**d**) Confocal laser scanning microscope images showing a section of the right (R) APG from the control (C) group. Arrows indicate FB+ neurons (**a**) immunoreactive simultaneously (**d**) against vasoactive intestinal polypeptide (VIP) (**c**) and dopamine-B-hydroxylase (DBH) (**b**). A small number of VIP-IR (**c**,**d**) nerve fibers are marked with arrowheads. The images were taken from blue, green and red fluorescent channels. Blue, green and red channels were digitally superimposed (**d**). Scale bar = 50 µm. (**e**–**h**) Confocal laser scanning microscope images showing a section of the right (R) APG from the experimental (E) group. The arrow indicates the FB+ neuron (**e**) immunoreactive with VIP (**g**) but not containing DBH (**f**). A large amount of DBH-IR (**f**) and VIP-IR (**g**) fibers and single DBH/VIP -IR (h) fibers marked with arrowheads. The images were taken from blue, green and red fluorescent channels. Blue, green and red channels were digitally superimposed (**h**). Scale bar = 50 µm.

**Figure 5 ijms-22-02231-f005:**
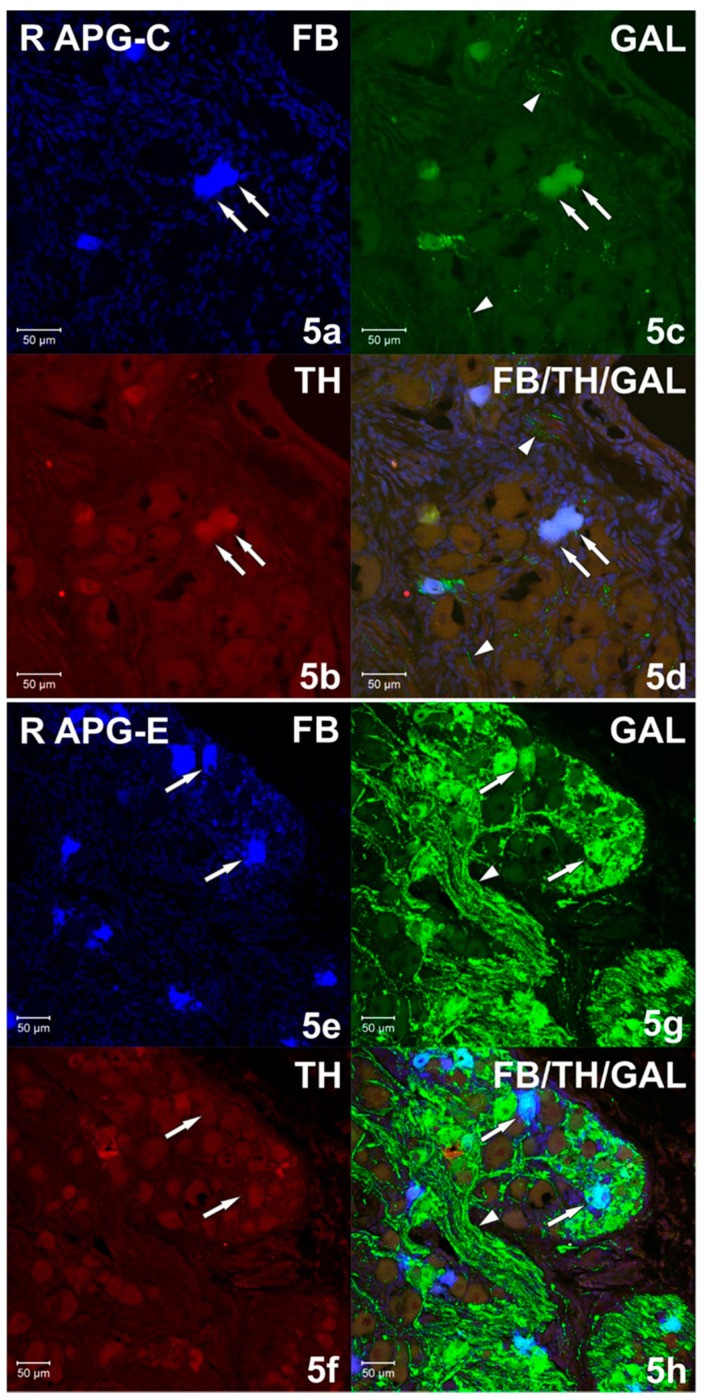
(**a**–**d**) Confocal laser scanning microscope images showing a section of the right (R) APG from the control (C) group. FB+ (**a**) neurons simultaneously immunoreactive (**d**) against TH (**b**) and galanin (GAL) (**c**) marked with arrows. The arrows indicate single, varicose GAL-IR fibers (**c**,**d**). The images were taken from blue, green and red fluorescent channels. Blue, green and red channels were digitally superimposed. Scale bar = 50 µm. (**e**–**h**) Confocal laser scanning microscope images showing a section of the right (R) APG from the experimental (E) group. Arrows indicate FB+ (**e**) GAL-positive (**g**) neurons, but no TH-IR (**f**). Very numerous GAL-IR fibers (**g**,**h**) marked with an arrowhead. The images were taken from blue, green and red fluorescent channels. Blue, green and red channels were digitally superimposed. Scale bar = 50 µm.

**Figure 6 ijms-22-02231-f006:**
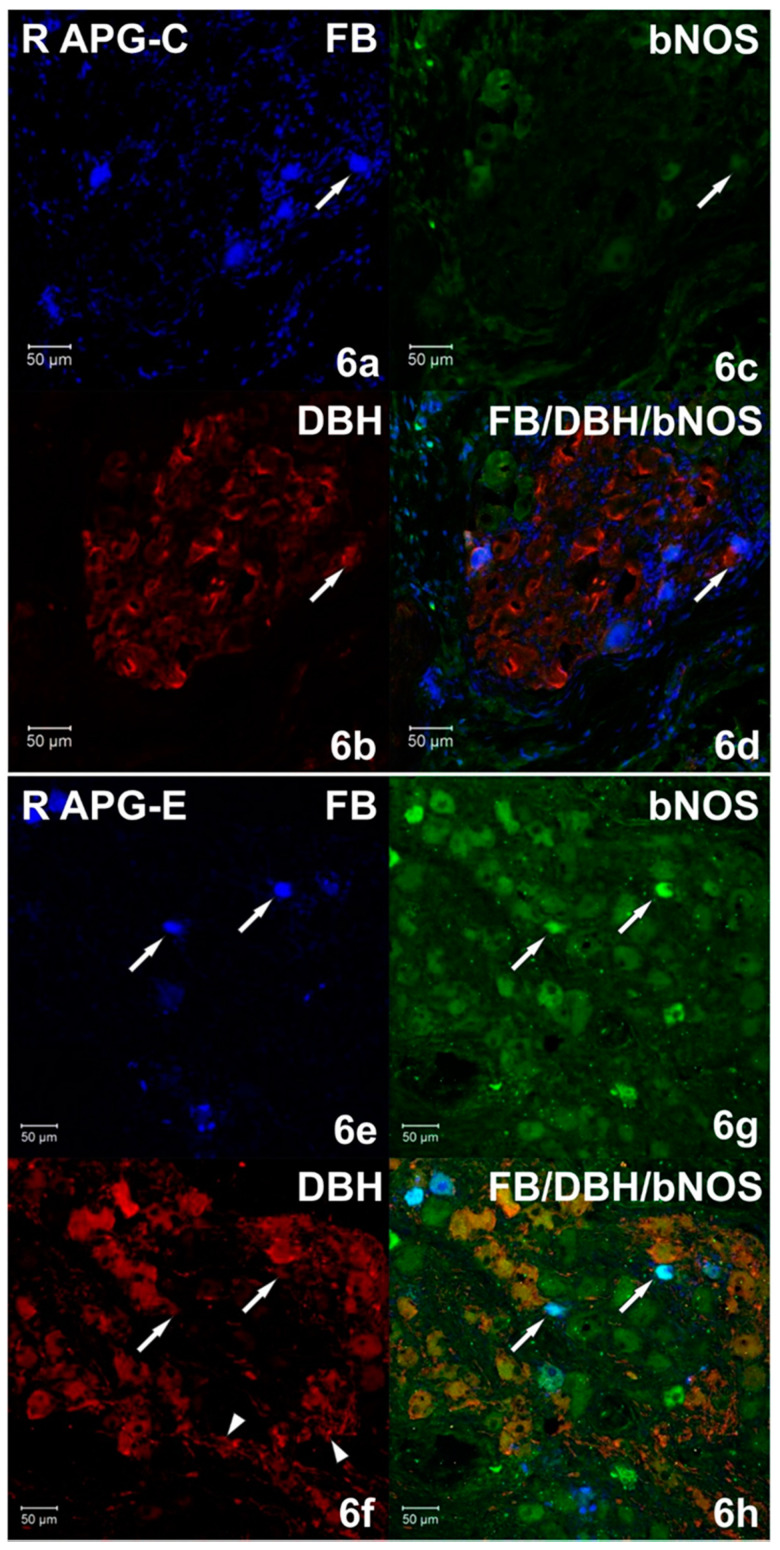
(**a**–**d**) Confocal laser scanning microscope images showing a section of the right (R) APG from the control (C) group. FB+ neurons (**a**) double-stained (**d**) for DBH (**b**) and brain nitric oxide synthase (bNOS) (**c**). The arrow marks an FB+ cell containing DBH but not bNOS. The images were taken from blue, green and red fluorescent channels. Blue, green and red channels were digitally superimposed. Scale bar = 50 µm. (**e**–**h**) Confocal laser scanning microscope images showing a section of the right (R) APG from the experimental group. Arrows indicate the FB+ (**e**) immunoreactive neurons against bNOS (**g**) and those not containing DBH (**f**). The arrowheads indicate a large number of DBH-IR (**f**) fibers. The images were taken from blue, green and red fluorescent channels. Blue, green and red channels were digitally superimposed. Scale bar = 50 µm.

**Figure 7 ijms-22-02231-f007:**
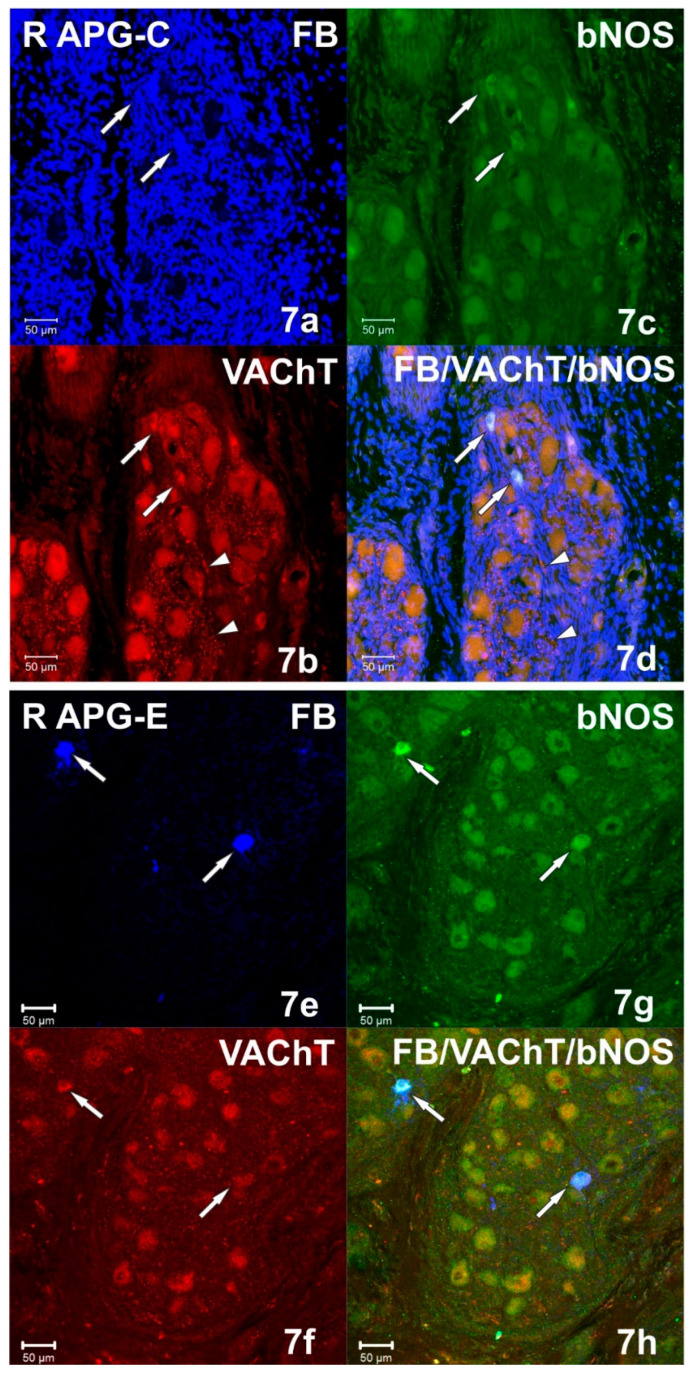
(**a**–**d**) Confocal laser scanning microscope images showing a section of the right (R) APG from the control (C) group. Arrows indicate FB+ (**a**) vesicular acetylcholine transporter (VAChT)-IR (**b**) neurons, but not containing bNOS (**c**). The arrowheads indicate numerous VAChT-IR fibers with a transverse course (**b**,**d**). The images were taken from blue, green and red fluorescent channels. Blue, green and red channels were digitally superimposed. Scale bar = 50 µm. (**e**–**h**) Confocal laser scanning microscope images showing a section of the right (R) APG from the experimental group. Arrows indicate FB+ neurons (**e**) immunoreactive against bNOS (**g**) but VAChT negative (**f**). The images were taken from blue, green and red fluorescent channels. Blue, green and red channels were digitally superimposed. Scale bar = 50 µm.

**Figure 8 ijms-22-02231-f008:**
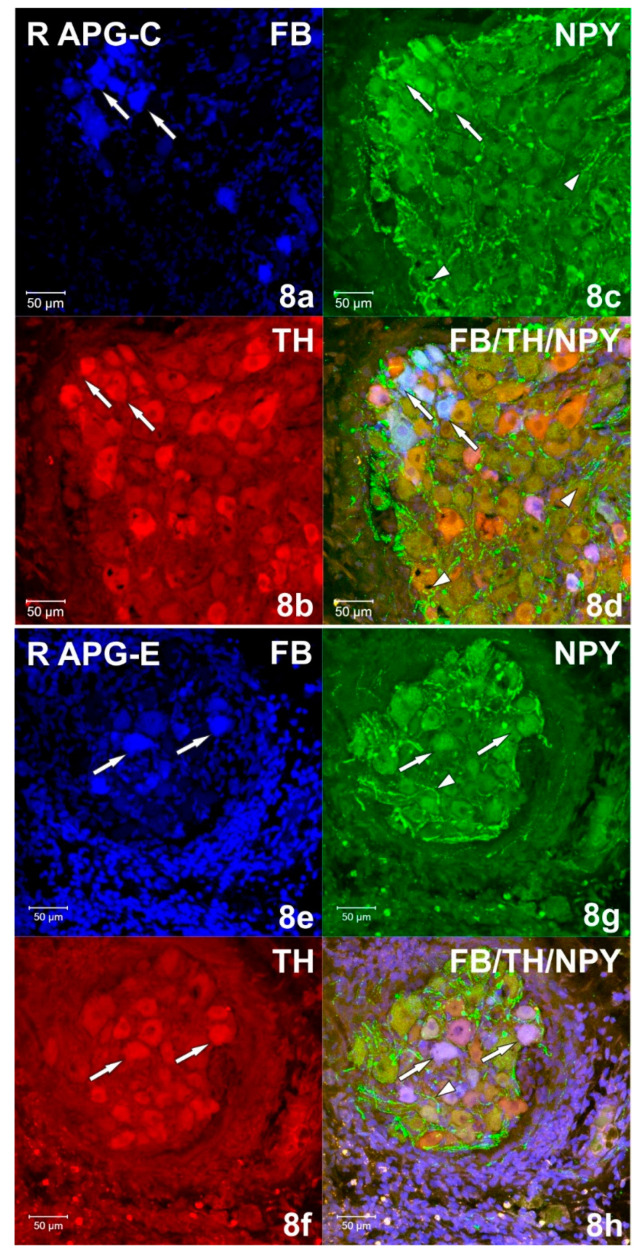
(**a**–**d**) Confocal laser scanning microscope images showing a section of the right (R) APG from the control (C) group. FB+ (**a**) neurons simultaneously immunoreactive (**d**) against TH (**b**) and NPY (**c**) marked with arrows. The arrows indicate single, varicose NPY-IR fibers (**c**,**d**). The images were taken from blue, green and red fluorescent channels. Blue, green and red channels were digitally superimposed. Scale bar = 50 µm. (**e**–**h**) Confocal laser scanning microscope images showing a section of the right (R) APG from the experimental (E) group. Arrows indicate FB+ (**e**) TH-positive (**g**) neurons and NPY-IR (**f**). Numerous NPY-IR fibers (**g**,**h**) marked with an arrowhead. The images were taken from blue, green and red fluorescent channels. Blue, green and red channels were digitally superimposed. Scale bar = 50 µm.

**Figure 9 ijms-22-02231-f009:**
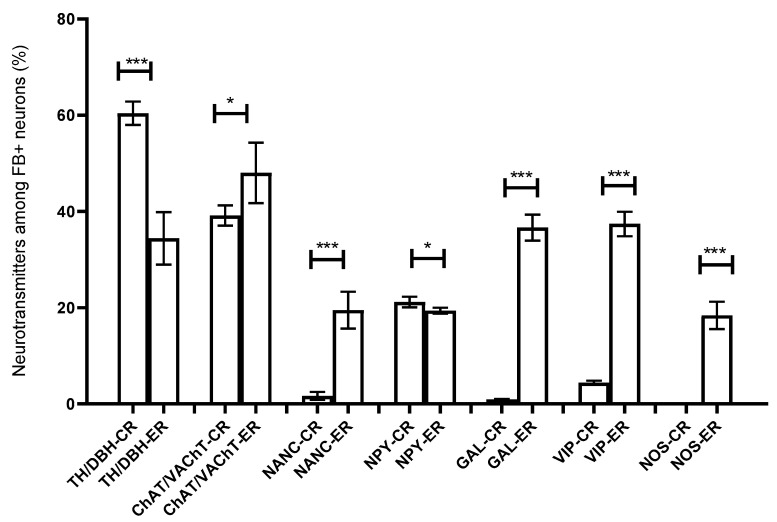
Percentage of FB+ neurons (±SEM) showing immunoreactivity against the tested substances located in the right (R) APG of the control (C) and experimental animals (E). * *p* < 0.05, *** *p* < 0.001.

**Figure 10 ijms-22-02231-f010:**
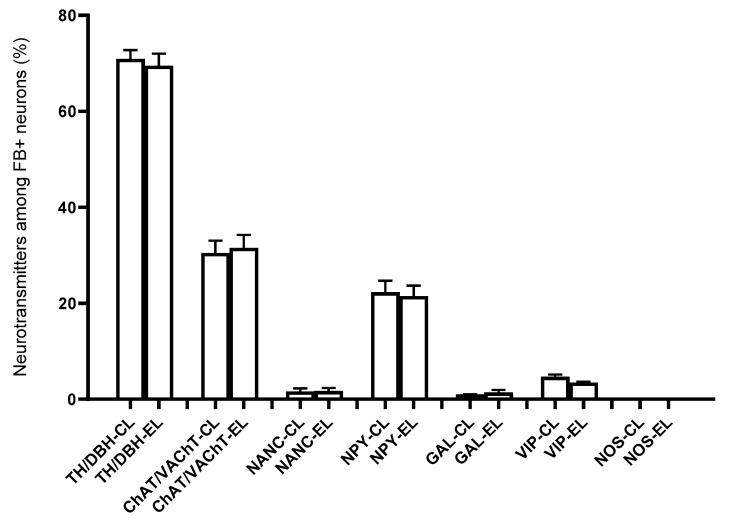
Percentage of FB+ neurons (±SEM) showing immunoreactivity against the tested substances located in the left (L) APG of the control (C) and experimental animals (E); There were no non-statistically (ns) significant differences (*p* > 0.05).

**Figure 11 ijms-22-02231-f011:**
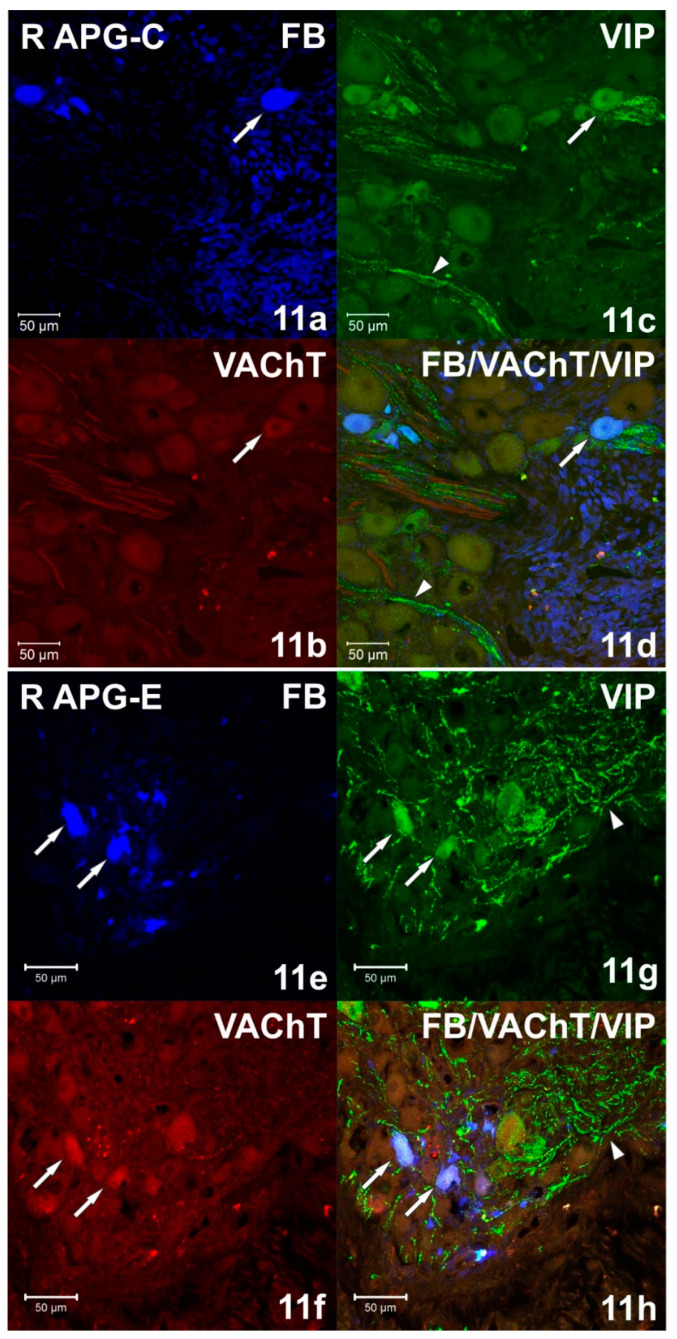
(**a**–**d**) Confocal laser scanning microscope images showing a section of the right (R) APG from the control (C) group. The arrow indicates the FB+ neuron (**a**) containing VIP (**c**) but VAChT negative (**b**). The arrowhead indicates the low number of VIP-IR fibers (**c**,**d**). The images were taken from blue, green and red fluorescent channels. Blue, green and red channels were digitally superimposed. Scale bar = 50 µm. (**e**–**h**) Confocal laser scanning microscope images showing a section of the right (R) APG from the experimental group. Arrows indicate FB+ neurons (**e**) simultaneously immunoreactive (**h**) against VAChT (**f**) and VIP (**g**). A large number of VIP-IR fibers (**g**,**h**) were marked with the arrowhead. The images were taken from blue, green and red fluorescent channels. Blue, green and red channels were digitally superimposed. Scale bar = 50 µm.

**Figure 12 ijms-22-02231-f012:**
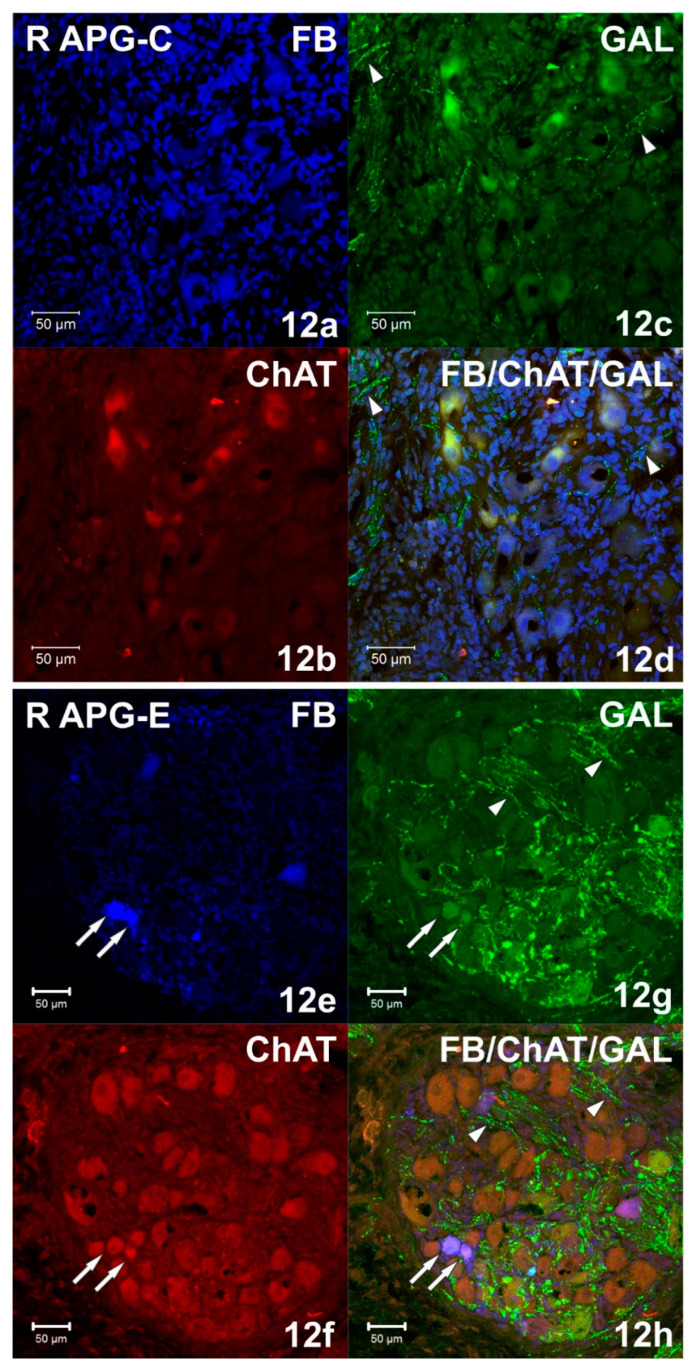
(**a**–**d**) Confocal laser scanning microscope images showing the right (R) APG section from the control (C) group. FB+ (**a**) neurons double-stained (**d**) for ChAT (**b**) and GAL (**c**). Single, varicose GAL-IR fibers (**c**,**d**) are marked with arrowheads. The images were taken from blue, green and red fluorescent channels. Blue, green and red channels were digitally superimposed. Scale bar = 50 µm. (**e**–**h**) Confocal laser scanning microscope images showing a section of the right (R) APG from the experimental (E) group. Arrows indicate FB+ (**e**) neurons immunoreactive with GAL (**g**) but not containing ChAT (**f**). A large number of GAL-IR varicose fibers (**g**,**h**) were marked with arrowheads. The images were taken from blue, green and red fluorescent channels. Blue, green and red channels were digitally superimposed. Scale bar = 50 µm.

**Figure 13 ijms-22-02231-f013:**
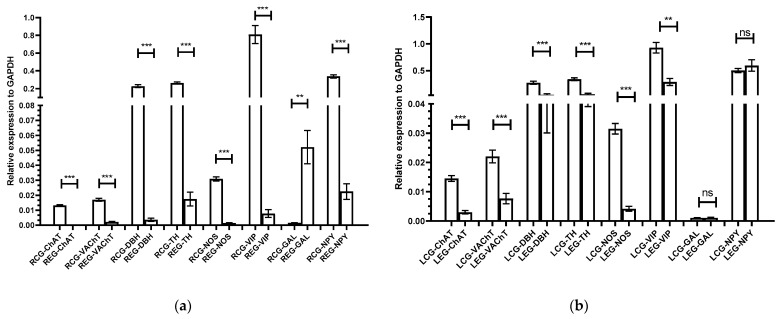
**a**,**b**. Analysis of the relative gene expression for ChAT, VAChT, DBH, TH, NOS, VIP, GAL and NPY in the right (R) (**a**) and left (L) (**b**) APG of the control (CG) and experimental (EG) group of animals, normalized against the GAPDH reference gene. *** *p* < 0.001, ** *p* < 0.01, ns—no statistically significant differences (*p* > 0.05).

**Table 1 ijms-22-02231-t001:** Antisera used in the study.

***Primary Antibodies***					
**Antigen**	**Clonality**	**Host**	**Dilution**	**Company**	**Catalog No.**
Dopamine beta hydroxylase -DBH	polyclonal	rabbit	1:200	Enzo	BML-DZ1020-0050
Tyrosine hydroxylase -TH	monoclonal	mouse	1:1000	Sigma Aldrich	T2928
Vesicular acetylcholine transporter -VAChT	polyclonal	rabbit	1:4000	Sigma Aldrich	V5387
Choline acetyltransferase -ChAT	polyclonal	goat	1:10000	Merk Millipore	AB144P
Galanin -GAL	polyclonal	rabbit	1:2000	Sigma	AB2233
Neuropeptide Y -NPY	monoclonal	rat	1:400	Enzo Life Sciences	BML-NA 1233
Nitric oxide synthase brain -bNOS	monoclonal	mouse	1:200	Sigma Aldrich	N2280
Vasoactive intestinal polypeptide	polyclonal	mouse	1:500	Biogenesis	9535-0504
***Secondary Antibodies***					
**Antigen**	**Fluorophore**	**Host**	**Dilution**	**Company**	**Catalog No.**
Mouse IgG	Alexa 488	goat	1:1000	Invitrogen	A-11001
Rat IgG	Alexa 488	goat	1:1000	Invitrogen	A-11006
Rabbit IgG	Alexa 488	goat	1:1000	Invitrogen	A-11008
Mouse IgG	Alexa 555	goat	1:1000	Invitrogen	A-21127
Rabbit IgG	Alexa 555	goat	1:1000	Life Technologies	A 21428
Goat IgG	Alexa 568	donkey	1:500	Invitrogen	A-11057

**Table 2 ijms-22-02231-t002:** Primers used in the study.

Gene	Forward Sequence 5’-3’	Reverse Sequence 5’-3’	Accession Number
*gapdh*	GATCGTCAGCAATGCCTCCT	GATGCCGAAGTGGTCATGGA	NM_001206359.1
*th*	TGCACCCAGTAYATCCGCCAYGC	TAGYTCCTGAGCTTGTCCTT	NM_214131.1
*dbh*	AACTCGTTCAACCGCCAAGT	AGGTTCCACTCACCCTGGAA	XM_001927211.6
*slc18a1 - VAChT*	CAACCTCTTTGGCGTGTTGG	ATGAGCCAGAGGCACACAAA	XM_021072276.1 XM_021072275.1 XM_021072274.1 XM_013990243.2
*chat*	GCTAGCCTTCTACAGGCTCC	AGTGGCCGATCGAATGTTGT	NM_001102473.1 NM_001104955.1
*gal*	TGGGCCACATGCCATCGACA	CGGCCTGGCTTCGTCTTCGG	NM_214234.1
*N nos*	TTCGTGCGTCTCCACACCAA	AGTACTTGAAGGCCTGGAAGA	XM_003357447.5 XM_005667633.3 XM_003130163.6 XM_013979715.2 XM_003130164.6
*vip*	CTGACAACTACACCCGCCTT	TCTCCTTCAATGCTCCGCTT	NM_001195233.1
*npy*	TCACCAGGCAGAGATACGGA	ACACAGAAGGGTCTTCGAGC	NM_001256367.1

## Data Availability

The data presented in this study are available in Appendix A here as S1.

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
