# Peer review of "Changes in the Neurochemical Coding of the Anterior Pelvic Ganglion Neurons Supplying the Male Pig Urinary Bladder Trigone after One-Sided Axotomy of Their Nerve Fibers"

_ijms, 2021, doi:10.3390/ijms22052231_

Round 1

Reviewer 1 Report

Major comments to the authors:

  • some spelling and grammar errors are present and Englis has to be revised.

  • Line 30: The authors can substitute “The autonomic nervous system, playing in accord with other parts of the nervous system, plays a crucial role in maintaining normal bladder function.” With “The autonomic nervous system, cooperating with other parts of the nervous system, plays a crucial role in maintaining normal bladder function.”. Moreover, to which "other parts of the nervous system " do the authors refer?

  • The authors report also modification in the the number and distribution of nerve cells and fibers in the left APG, but no images are showed. Being a immunohistochemical work it could be usefull show significant images also for the left APG.

  • The authors underline also changes in NPY cells and fibers immunoexpression, at list in the right APG. Why they dont show images of NPY-immunoexpression?

  • The VAChT-immunoexpression in the control sample in fig. 7 seems very different from what the authors show in the Fig. 10.

Author Response

Response to Reviewer 1:

Comments and Suggestions for Authors

Major comments to the authors:

  • some spelling and grammar errors are present and English has to be revised.

The language was revised by two linguistic editors.

  • Line 30: The authors can substitute “The autonomic nervous system, playing in accord with other parts of the nervous system, plays a crucial role in maintaining normal bladder function.” With “The autonomic nervous system, cooperating with other parts of the nervous system, plays a crucial role in maintaining normal bladder function.”. Moreover, to which "other parts of the nervous system " do the authors refer?

The mentioned fragment of text was removed as we have found it being of little relevance to the Introduction.

  • The authors report also modification in the number and distribution of nerve cells and fibers in the left APG, but no images are showed. Being a immunohistochemical work it could be useful show significant images also for the left APG.

One of referees demanded the reduction of the number of figures. We tried to keep the number of figures low, yet still informative. However, we showed additionally the distribution of FB+ neurons in the left APG (control and experimental animals) modifying Fig. 2).

  • The authors underline also changes in NPY cells and fibers immunoexpression, at list in the right APG. Why they dont show images of NPY-immunoexpression?

 We added Fig. 8 showing the example of immunohistochemical staining for NPY combined with TH in the right control and experimental APG.

  • The VAChT-immunoexpression in the control sample in fig. 7 seems very different from what the authors show in the Fig. 10.

We found more appropriate set of pictures reflecting the described morphology.

Reviewer 2 Report

In this article, the authors use UBT to investigate FB+ neuron and condition distribution with PCR and immunostaining, the conclusions are clearly presented. A couple of suggestions, the wording of the article should be modified a bit for English style, the titles of the figures can be consolidated if they are from the same figure numbers. Finally, could they author quantify immunostaining results and compare with PCR to see if they are nicely correlated?  

Author Response

Response to Reviewer 2:

Comments and Suggestions for Authors

“In this article, the authors use UBT to investigate FB+ neuron and condition distribution with PCR and immunostaining, the conclusions are clearly presented. A couple of suggestions, the wording of the article should be modified a bit for English style, the titles of the figures can be consolidated if they are from the same figure numbers. Finally, could they author quantify immunostaining results and compare with PCR to see if they are nicely correlated?”

The language was reedited by two linguistic editors. As regards quantification of immunostaining and relating it to qPCR results it is necessary to mention that the quantification of immunostaining is practically impossible since we cannot prepare the standard curve, the linearity of immunostaining intensity is unpredictable and the immunohistochemical staining intensity is dependent on many variables, like the fixation of tissue. The quantitation of immunostaining using, for example, image processing software would give nice numbers of no relevance.

Reviewer 3 Report

The authors reported that the results of various marker expressions on the tissue and mRNA in one-sided axotomy of nerve fibers using the pig, and suggested neuronal plasticity under pathological condition. However, the results are only histology and gene expression, thus it is over-discussed. There are several concerns against the results they showed. The comments on the manuscript are as follows; 

  1. It is necessary to examine the cell death, since the change in the neuronal population due to axotomy could be attributed to cell death.
  2. The qPCR results seem to show relative expressions, but I don't know what the relative expressions for. GAPDH should be used for normalization, and not for relative expression. Also, there is misspelling “exspression”.
  3. Overall, there were too many figures, which were very difficult to understand and connect the results from the figure legends. For example, qPCR results should present together as a one big figure with a meaningful figure legend.
  4. Datasheets for many commercial antibodies do not list specificity for pigs. Concerns remain, as there are no experimental results to show that the staining signals are specific.
  5. I could not get the meaning of “plasticity”, because one-sided axotomy affects of course affects the neuronal population. There were no morphological data as well as neuronal activity data such as electrophysiology. Therefore, it was difficult to get the meaning of “plasticity”.
  6. The discussion is over-discussion compared to the results obtained.

Author Response

Response to Reviewer 3:

Comments and Suggestions for Authors

The authors reported that the results of various marker expressions on the tissue and mRNA in one-sided axotomy of nerve fibers using the pig, and suggested neuronal plasticity under pathological condition. However, the results are only histology and gene expression, thus it is over-discussed. There are several concerns against the results they showed. The comments on the manuscript are as follows; 

Yes, we suggested the plasticity of the nerve cells under pathological condition to show their possible importance for clinical sciences. Of course, it is a kind of rhetoric figure and it was skipped from the Discussion, along with some other fragments of text we found to be of little relevance, to make it more concise.

  1. It is necessary to examine the cell death, since the change in the neuronal population due to axotomy could be attributed to cell death.

In general, we were not interested in cell death in this experiment. Of course, it is possible and desirable to study multiple aspects, but we had to narrow our interests to make the project manageable. However, we have done stainings for caspase-3 in sections we saved for future studies. We did not find any apparent differences in the immunoexpression of caspase-3 in FB+ neurons between the control and experimental animals. We include the figure for your reference showing caspase-3 staining in the control and experimental APG. We do not include the figure into the paper since it would need to alter the Discussion and include lengthy text discussing the aspects of cellular death.

2. The qPCR results seem to show relative expressions, but I don't know what the relative expressions for. GAPDH should be used for normalization, and not for relative expression. Also, there is misspelling “exspression”.

Misspelling was corrected. In order to dispel the Referee’s doubts about the housekeeping gene considered in the qPCR investigations, we would like to emphasize that in our molecular biology laboratory we always test at least two housekeeping genes for the tissues studied (e.g. Zalecki et al., Neurogastroenterol Motil, 2018, 30(7):e13360), and finally choose the most appropriate one for further investigations. In case of the present study, the GAPDH appeared to be the most suitable. Of course, GAPDH was used for normalization of qPCR results and it is now clearly stated in the text.

3. Overall, there were too many figures, which were very difficult to understand and connect the results from the figure legends. For example, qPCR results should present together as a one big figure with a meaningful figure legend.

The qPCR results were shown on separate figures due to very large differences in the expression level making it impossible to show meaningfully in one graph. However, we found the way to solve the issue you have addressed making a discontinuous graph accommodating data with both very low and very high values. Now the qPCR results figures are combined into one figure. The figures we show in the paper are kept to minimum from the point of view of the paper concerning the morphology. One of the referees demanded that the number of figures in increased to show additional data important from the point of view of morphology and immunohistochemistry. We think that the number of figures we show is a good compromise between informativeness and conciseness.

4. Datasheets for many commercial antibodies do not list specificity for pigs. Concerns remain, as there are no experimental results to show that the staining signals are specific.

The antibodies used in the study were applied by us to the pig tissues many times and were thoroughly checked for specificity. We tested these antibodies previously for specificity with preabsorbtion tests and we include omission and replacement tests into each experiment, which was not mentioned by mistake. Now, the relevant information about specificity tests is introduced into the text of the paper.

5. I could not get the meaning of “plasticity”, because one-sided axotomy affects of course affects the neuronal population. There were no morphological data as well as neuronal activity data such as electrophysiology. Therefore, it was difficult to get the meaning of “plasticity”.

We changed the term “plasticity” into “changes in the neurochemical coding” which more appropriately reflects the changes we were investigating.

  1. The discussion is over-discussion compared to the results obtained.

We shortened the discussion to make it more concise.

Round 2

Reviewer 1 Report

The authors have corrected and substantially improved the manuscript as requested

Reviewer 3 Report

The authors correctly responded to the reviewer’s comments and revised manuscript appropriately. I can agree the publication of revised version of manuscript. I have no further comment.